# Balancing between being the most valuable player (MVP) and passing the ball: a qualitative study of support when living with chronic pain in Sweden

Veronica Lilja ,[1,2] Sara Wallström ,[1,2,3,4] Markus Saarijärvi ,[2,5,6] Mari Lundberg,[2,7] Vivi-Anne Segertoft ,[2] Inger Ekman[1,2,8]

For numbered affiliations see end of article.

**Correspondence to**
Veronica Lilja;
veronica.lilja@gu.se

## ABSTRACT

**Objective** This study aimed to elucidate the meaning of lived experiences of support from social networks and the healthcare sector in persons with chronic pain.

**Design** A qualitative, phenomenological hermeneutic method was used to analyse interview data.

**Setting** Participants were recruited from patient organisations in Sweden.

**Participants** Ten (seven women, two men and one non-binary) individuals with chronic musculoskeletal pain were included.

**Findings** The meaning of lived experiences of support in persons with chronic pain involves balancing between being the most valuable player (MVP) and passing the ball, meaning balancing between being a capable person and accepting support to be that capable person.

**Conclusion** For participants who lived with chronic pain, support means balancing between being capable (the MVP) and willing to accept support (passing the ball), which aligns with the concept of person-centred care. Our findings may be useful for policy-makers, managers and clinical professionals when planning and performing care for persons with chronic pain. Future research should focus on how the healthcare sector can create support to enable persons with chronic pain to be the MVP while being able to pass the ball to their social networks and the healthcare sector.

## STRENGTHS AND LIMITATIONS OF THIS STUDY

⇒ Chronic pain affects many people worldwide and understanding the meaning of the healthcare sector and social network support is vital to provide tailored assistance and practical solutions.
⇒ Further insights were achieved with a patient representative actively involved in the analysis and manuscript preparation.
⇒ A diversity of basic demographic indicators (age, geographical location and occupational status) is a strength of the study.
⇒ A limitation is that most participants had post high school education, were female (all eligible participants with other genders were included) and were born in Sweden.

## INTRODUCTION

Pain is defined as 'an unpleasant sensory and emotional experience associated with, or resembling that associated with, actual or potential tissue damage'.[1] Chronic pain persists or recurs for over 3 months and is classified as a disease on its own, not just a symptom.[2] The prevalence of chronic pain differs between studies, contexts and types of measurement. In a large European study comprising 16 countries, the prevalence of chronic pain was estimated to be 19%.[3] A US study showed a prevalence rate of 20.4%.[4]

Persons with chronic pain often have comorbidities (eg, depression, anxiety, cardiovascular disease and cancer),[5] side effects from medication[6] and poor health-related quality of life.[7] Chronic pain can adversely affect sleep, daily activities, relationships and the ability to work.[6] Pain is often perceived as invisible to others, which can contribute to feeling unjustly treated in society.[8] From a societal and health economic perspective, chronic pain presents challenges because it is a common reason for sick leave[3 9 10] and healthcare-seeking behaviour.[5 6 9] In Sweden, the cost (indirect and direct) of chronic pain-related diagnoses was estimated at €32 billion per year in 2012, of which 59% were due to sick leave and early retirement.[11]

A meta-synthesis showed that support from family and friends is important in pain management.[12] Social support can include sharing advice, expressions of empathy and contributing to positive feelings.[13 14] In contrast, lacking support can lead to feelings of loneliness and not being needed.[15] Peer

support interventions have been shown to decrease pain severity and interference.[16] However, there is conflicting evidence of the positive effects of support. A peer support intervention for veterans with musculoskeletal pain found no statistically significant impact on pain.[17] Studies investigating spouses' participation in educational interventions suggest no additional benefits of including a partner[18] and that participating with a partner could make participants more prone to fatigue and lower self-efficacy compared with not participating with a partner.[19]

Collaborative relationships with healthcare professionals constitute support that facilitates self-management of pain.[12] The biopsychosocial model and the multimodal approach have been shown to improve pain management.[20 21] However, it has also been reported that persons with chronic pain feel that healthcare professionals rarely take their condition seriously and that they desire better support from health professionals.[3 15 22]

Due to conflicting evidence and the complexity of support for persons with chronic pain, there is a need to understand the meaning of support, both within and outside the healthcare system. A deeper understanding of the phenomenon would facilitate the comprehension of the need for support and could aid in bringing clarity on what kind of support persons with chronic pain want and need.

Therefore, this study aims to elucidate the meaning of lived experience of support from social networks and the healthcare sector in persons with chronic pain.

## METHODS
### Design
The present study applied a qualitative method with a phenomenological hermeneutic approach inspired by Lindseth and Norberg.[23] Phenomenological hermeneutics is suitable for interpreting the essential meaning of a lived phenomenon through text narratives.[23] This study follows the Standards for Reporting Qualitative Research guidelines.[24]

### Participants and setting
Participants were recruited from four Swedish patient organisations by a Facebook post or an email sent from the organisations. These organisations are well established, with members having different diagnoses and many having chronic pain in common. Persons willing to participate in the study contacted the first author (VL) by email. Inclusion criteria were ≥18 years of age, living in Sweden and having chronic musculoskeletal pain (defined as 'chronic pain arising from musculoskeletal structures'[25]). Persons who primarily seemed to struggle with other conditions and congenital diseases, such as concurrent cancer diagnosis, were excluded. Participants who mainly wanted to share their narratives about musculoskeletal pain but had congenital diseases, undergone cancer treatment or had another pain-related diagnosis were not excluded.

Some 177 persons (1 non-binary, 4 men, 172 women) expressed interest in participating. A purposive sampling strategy was employed to include participants from different parts of Sweden, regardless of treatment or current contact with healthcare. Five participants mentioned receiving support from the healthcare sector and social networks by starting to share their narratives in the email expressing interest in participating. They were purposefully selected as they were willing to share their vast experience of the phenomenon under study, allowing the collection of rich data.[26] Maximum variation sampling allowed the discovery of common meanings across demographic differences.[26] Therefore, a diversity of experiences of support, age, geographical location, sick leave rate and background diagnosis was sought even though most potential participants were women. Eight participants were initially included, and their narratives were deemed sufficient to answer the study's research question. Another two participants were interviewed to achieve greater variation in education level. None of the participants declined to participate. The material was considered rich enough to find meanings of support. After discussions in the research group, inclusion was halted at 10 participants.

The author who conducted the interviews (VL) has a nursing and public health background. Before the study, the interviewer's preunderstanding was written (see online supplemental file 1) and reflected on in the analysis. VL was a novice in phenomenological hermeneutics; however, the research group's extensive experience complemented her lack of practice in this field. A patient representative was also part of the research group and contributed with experience of living with chronic pain.

### Data collection
A semistructured interview guide (online supplemental file 2), derived from Brinkmann and Kvale,[27] was developed by VL with input from SW, ML and IE. The guide included three domains of support: the healthcare sector, social networks and how support from social networks could be integrated into care. The three domains were chosen based on their previously described importance.[3 12–16 19–22] The interview guide contained open-ended questions with suggestions for additional probing questions. It was piloted in the first two interviews and revised by changing from the question 'Which persons outside of the healthcare sector have you gotten support/help from?' to 'Which persons outside of the healthcare sector have been important to you regarding your pain?' The final version of the question better facilitated narratives about social networks. The narratives from the two pilot interviews were deemed relevant, as they answered the research question and were thus included in the data analysis. Seven interviews were conducted digitally through Zoom, one by phone, and two face-to-face between February 2021 and August 2022. The interviews were recorded and transcribed verbatim by VL. Participants chose the interview date, place and format. Video

**Table 1** Examples of structural analysis

| Text | Condensed meaning units | Subtheme | Theme | Main theme |
|---|---|---|---|---|
| The appreciation of myself was destroyed already at the beginning of my sickness … and the value… my own value. Because they did not see me as competent. They saw only my illness. So, this has been awfully hard. And there is still frustration in not getting this kind of recognition. | Self-worth | Being a valuable player and not only being the injured one. | Being the MVP | Balancing between being the MVP and passing the ball in a match against pain |
| I was so incredibly fortunate; he is a great guy, a great doctor and, and… he supports me in the next step. | Someone fights with you | Being able to pass the ball when you have to | Passing the ball | Balancing between being the MVP and passing the ball in a match against pain |

MVP, most valuable player.

and telephone interviews are trustworthy alternatives to face-to-face interviews in qualitative research.[28]

## Patient and public involvement
Four patient organisations contributed to recruiting study participants and will contribute to disseminating the findings. One of the coauthors (V-AS) is a patient representative. V-AS actively participated in data analysis and manuscript preparation.

## Data analysis
Data were analysed with phenomenological hermeneutics. The method, influenced by Ricoeur's theory of interpretation and developed by Lindseth and Norberg,[23] involves three intertwined phases: naïve understanding, structural analysis and interpretation of the whole. Through the hermeneutic spiral, the phases are constantly overlapping, revisited and compared with each other to move between explanation and understanding by interpretation of the whole and the parts.[29]

### Naïve understanding
Each interview was read several times. VL formulated a naïve understanding for each interview before constructing a merged naïve understanding of all interviews.

### Structural analyses
The structural analyses (see table 1) were performed with the software NVivo V.12 by VL with input from the other authors. The text of each interview was divided into meaning units and condensed. All text was considered, but only text associated with the study's aim was included in the structural analyses. The condensed meaning units were continuously compared with the naïve understanding. The interviews were read through again, and the naïve understanding was revised and compared with the structural analyses. This process was repeated several times. Eventually, tentative themes and subthemes were formulated and compared with the condensed meaning units and the naïve understanding. VL and IE

continuously discussed and reformulated the tentative findings before consulting the other authors.

### Interpretation of the whole
In interpreting the whole VL and IE compared the preunderstanding, the naïve understanding and the structural analyses several times to identify inconsistencies. The analysis was revised until all parts were consistent. The underlying meaning in the data was reflected on and compared with the existing literature, such as previous studies, the work of the philosopher Ricoeur and the underpinnings of person-centred care, yielding a new understanding. ML read all the interviews and the findings to ensure the interpretations were reasonable before giving feedback on the naïve understanding, structural analyses and interpretation of the whole. The understanding of the meaning of the findings was discussed among all authors several times to connect their perspectives, knowledge and understandings. The interpreted metaphor was developed through discussions among all authors based on the link to the naïve understanding and structural analyses. When consensus on the meaning of the findings and the metaphor was reached, the interpretation of the whole was formulated.

## FINDINGS
Ten participants were included in the study (two men, one non-binary and seven women). All eligible participants of other genders than female were included. The demographic characteristics of participants are described in table 2. The interviews lasted between 39 and 101 min (mean 77 min). Findings from the analyses are presented in the following order: the naïve understanding, the structural analyses and the interpretation of the whole.

### Naïve understanding
The naïve understanding of the meaning of support is that it reinforces participants' ability to manage their pain and everyday life. Participants seek to address their pain and life situation independently but need support to

**Table 2** Participant characteristics

| Characteristics | |
|---|---|
| **Age** | |
| Mean (range) | 48 years (24–70) |
| **Sex** | |
| Female | 7 |
| Male | 2 |
| Non-binary | 1 |
| **Place of birth** | |
| Born in Sweden | 9 |
| Not born in Sweden | 1 |
| **Diagnosis** | |
| Fibromyalgia | 1 |
| Postpolio syndrome | 2 |
| Spinal injury | 1 |
| Ehlers-Danlos syndrome (hypermobile Ehlers-Danlos Syndrome and Hypermobility Spectrum Disorder) | 4 |
| Fibromyalgia and Ehlers-Danlos Syndrome | 2 |
| **Duration of pain** | |
| Mean (range) | 32.6 years (14–66) |
| **Education level** | |
| High school degree | 3 |
| University degree | 7 |
| **Occupational status** | |
| Working full time | 5 |
| Working part time due to sick leave | 3 |
| Unemployed without financial support | 1 |
| On disability pension | 1 |
| **Relationship status** | |
| Partner | 7 |
| Single | 3 |

achieve these goals. They feel lonely in and diminished by the healthcare sector, often seen as unavailable or hostile. Support from the right healthcare professional, someone who listens and will go the extra mile to establish a diagnosis and provide help, makes a big difference in participants' perception of their own capability. Experiences of support from outside the healthcare sector vary considerably. Although participants are eager to manage independently, social networks that believe in them, show compassion and fight together with them are essential.

## Structural analyses

The main theme, themes and subthemes are described below and in table 3. The metaphor of a football match is used throughout the designation of the main theme, themes and subthemes to elucidate the meaning of support. The metaphor is related to participants' narratives and elaborated on under each heading.

### Balancing between being the most valuable player and passing the ball in the match against pain

Chronic pain can be a constant battle, and just like a football match, it can vary in intensity. In this match, pain is the opponent, the person with chronic pain is the most valuable player (MVP), and the teammates are individuals within the MVP's social networks and healthcare sector. The social networks could include family, partners, friends, employers, colleagues, peers with chronic pain, personal trainers, personal coaches, persons performing complementary therapies, neighbours and pets. The ball (designated as pain management) in this football match is passed around to members of the MVP's team to win the match against pain. The attempt to win the match does not mean being pain-free but living the life the person with chronic pain wants to live despite the pain. The meaning of lived experiences of support is the constant balancing act between managing alone (being the MVP) and accepting help from others (passing the ball), which is further explained through the themes and sub-themes.

### *Being the MVP*

Participants wanted to contribute to society like everyone else and manage independently by taking the lead in their care and daily life. They also desired to be who they were without pain dominating their lives. Contemplating the metaphor, this can be interpreted as they aspired to be the MVP in all aspects of their lives. Being believed and listened to were important aspects of being trusted to dribble the ball, which is essential when seeking to be the MVP.

### Being a valuable player and not just the injured one

Participants sought to be recognised as the persons they were, with unique personalities and experiences, which could often be difficult to achieve. When perceived as a product of their pain, they felt excluded and viewed as someone who could not accomplish much. When the social networks provided support by accepting the pain as part of the participants but still recognising them for

**Table 3** Overview of the main theme, themes and subthemes

| Main theme: Balancing between being the most valuable player (MVP) and passing the ball in the match against pain | |
|---|---|
| Theme: Being the MVP | Theme: Passing the ball |
| Subtheme: Being a valuable player and not just the injured one | Subtheme: Being part of a team |
| Subtheme: Being trusted to dribble | Subtheme: Having teammates when you have been tackled |
| | Subtheme: Worrying about being a benchwarmer |

who they were and their capabilities, it facilitated their acceptance of themselves and the pain.

> They used to ask me,' Why are you limping?' I said,' I played football last weekend,' and I have never played football. So he [participant's partner] told me, 'Now you're going to stop to tell them that you've played football.' After that, I felt fine. There was no problem. I even have a colleague who says, 'You're limping. Maybe you should sit down.' I don't have to be perfect all the time. *Participant 4*

Pets could also contribute to this support, providing unconditional affection and companionship. Participants also wanted healthcare professionals to recognise them as the persons they were. They felt supported when healthcare professionals aimed at strengthening their resources by not only focusing on their limitations but also on their abilities. Participants felt an enhanced power by accessing self-help devices (eg, orthoses).

> It is about being an important part of society, to contribute… instead of being the one others should take care of. Self-help devices help to achieve that. *Participant 10*

The social networks provided support by requesting the participants' help and advice. This support affirmed the participants' view of themselves as unique, capable, meaningful and contributors rather than just someone with pain. Providing peer support to others with pain within a patient organisation exemplifies how participants contributed.

> I find good support in supporting others. You get a reflection of yourself that way. So, maybe that is my best support, to support others. […] It was perfect to have somebody who needed me. *Participant 3*

Being recognised as a person rather than someone struggling with pain could be interpreted as being considered a valuable player in football. Valuable football players are still useful to the team when injured, as everyone appreciates their efforts and knows their potential. They are not regarded as 'that injured player.' They are still valuable, and everyone is eager to see them return to the field.

### Being trusted to dribble

To participants, a diagnosis was important to have their pain experiences taken seriously, understanding their pain and being believed, but it was often perceived as a challenging process. A diagnosis meant validating their condition and was also experienced as facilitating being believed, trusted and understood by the social networks. Being believed, trusted and understood by others was a support, and it also encouraged that expectations from others did not clash with the participants' abilities. When participants were trusted with tasks they could perform, their view of themselves as capable persons was reinforced.

> I have to put on a mask in front of people and pretend to be happy. But my friend says, 'I don't mind that you're low and in pain. You don't need to be happy and energised. We can still have coffee.' It's like getting rid of a 20 kg backpack. He understands. *Participant 7*

Participants sometimes felt they received better support within complementary therapies compared with traditional care. They felt listened to and perceived as capable. This complimentary support, however, was not always affordable because of the participants' often strained economy.

> My massage therapist sees me. I think that is a need that everyone has, to be seen and listened to. *Participant 2*

Being believed and listened to when sharing experiences of pain could be viewed as a football player being trusted by other teammates. The teammates show that the player and their abilities are trusted by passing them the ball and encouraging them to dribble, meaning taking responsibility for the next move.

### Passing the ball

Living with pain was challenging, and occasionally, participants felt their abilities were inadequate. They needed a team to help them regain trust in themselves and fight to improve their situation. In football, the MVP must have a team to pass the ball to because they cannot win the game alone.

### Being part of a team

Participants often felt alone, struggling with pain. Navigating the healthcare system and the social insurance agency's policies was difficult. Fighting different systems alone evoked feelings of being diminished, vulnerable and powerless. When they found regular contact with a healthcare professional willing to help and possibly involve other professionals, or when their networks helped them fight the system, they did not feel as lonely. Regular contact enhanced the participants' sense of capability. Having support from others, they felt more confident and could accomplish more.

> My personal trainer said, 'I have a plan. Let's focus on this so you can continue fighting.' Thanks to her, I am stronger and I feel like things are moving forward. *Participant 1*

> I want to take responsibility for my pain and situation myself, but to be able to have someone to ask for help when I can't bear to deal with it. Like a backup, a support that stands on the side but does not overtake the main responsibility. It is very easy to end up in a subordinate position when you have a chronic condition, that you're in the hands of others. I want to be in charge but still have that support system around me as a backup for when I'm worse. When I do not

have that, I have to push myself beyond my limits, and my health deteriorates. *Participant 8*

Reducing loneliness by accepting support from others could be analogous to being part of a football team. The individual is stronger with teammates and has a better chance of beating the other team (ie, living the desired life despite pain). Being part of a team also means being included and not alone.

### Having teammates when you have been tackled

Participants felt hopeless when told by healthcare professionals there was nothing left to be done after undergoing several medical treatments or when test results failed to reveal the cause of their pain. The hope of feeling capable was restored when they were provided support (eg, tailored training programmes or self-help devices). When social networks provided support by facilitating daily tasks, tailored work schedules, ideas for pain management or contacts with trusted healthcare professionals, participants felt hope that their circumstances would improve, and they could live the desired life.

> I was advised to change my diet. It helped me so much. It somewhat improved my state because I could affect my situation a little bit myself. *Participant 5*

Regaining hope could mean having a teammate take the ball when the football player has been tackled and cannot dribble on their own (when the pain is unbearable, and they do not know how to move on). By having a teammate take the ball (pain management), the hope remains that the team can control the ball instead of losing it to the opposing team (the pain). There is hope that there is something else to try, even though all options for pain management are exhausted and someone is fighting with you when you feel like your abilities are not enough to fight the pain on your own.

> When I get this lumbago, I believe it will never pass, and I have to live my whole life like this. And then I talk to him [participant's partner], and he reminds me it will pass, and I'll be alright. It is good to be reminded. Otherwise, I'll go into that tunnel, thinking everything will go bad. *Participant 4*

### Worrying about being a benchwarmer

Participants found it challenging to determine what they could share about their situation with their networks (particularly family and friends) without appearing as a burden. Not wanting to strain their networks or cause worry discouraged participants from including them in their care.

> I don't know how much is okay to… share. What is oversharing? And how much can I share so it becomes enough, so they understand, but it does not become too much? It is a balancing act, and I find it difficult to know where the boundaries are. I dare not take that leap to tell them about my situation. *Participant 6*

Some healthcare professionals blamed participants for their situation, whereas others went beyond their regular duties to support participants by working during their free time (eg, lunch break). Such support made a huge difference, as participants needed their help. At the same time, the feeling of being a burden was enhanced because the healthcare professionals had to sacrifice their spare time to offer that support. Fear of being a burden due to pain could be seen as fear of losing the title of MVP and becoming a benchwarmer, that is, always on the bench without the opportunity to participate and contribute.

> You are sort of like a ball being kicked around within the system. And even if you try to tell them 'No, this is the field, this is the playing field', it becomes a bit like whatever. […] But I have been so lucky that these people are helping me out of pure, goodhearted will, even though they do not have time for it. Because their bosses tell them, 'You should only do this or that because these are our resources.' And I mean, they even helped me during lunch and such things. I must be careful when asking them for help because they are so kind-hearted. *Participant 9*

### Interpretation of the whole

As the main theme suggests, the meaning of the participants' lived experiences of support from the healthcare sector and social networks is to balance being a capable person and accepting help from others to continue being that capable person. Being capable means being recognised for who you are as a person and your qualities, to contribute, to be trusted and listened to. Accepting support means to not be alone, having someone fighting with you in order to enhance one's capability, but also to worry about being a burden. In football terms, being capable corresponds to being the MVP and accepting support corresponds to passing the ball. Developing the interpreted football metaphor; the MVP and the team compete against pain (see figure 1).

Many football players dream of being the MVP, whose actions ultimately determine the match's outcome. Our interpretation of the findings is that persons with chronic pain are no different, that is, they aspire to be important. However, not even the MVP can win a match without team support. They need the team to believe in their capability and support them to reach their full potential.

In accordance with how the MVP takes the lead, we would argue that persons with chronic pain take the lead in their daily lives and care by dribbling the ball (pain management). However, there must be a balance, as no team will win a match through only having the MVP dribbling, that is, it is also necessary to pass the ball. Passing the ball does not mean losing the title as MVP; instead, it means boosting the chances of winning the match through the help of teammates. Living with chronic pain often leads to accepting a life with pain. Aspiring to win the match against pain is not about being pain-free but

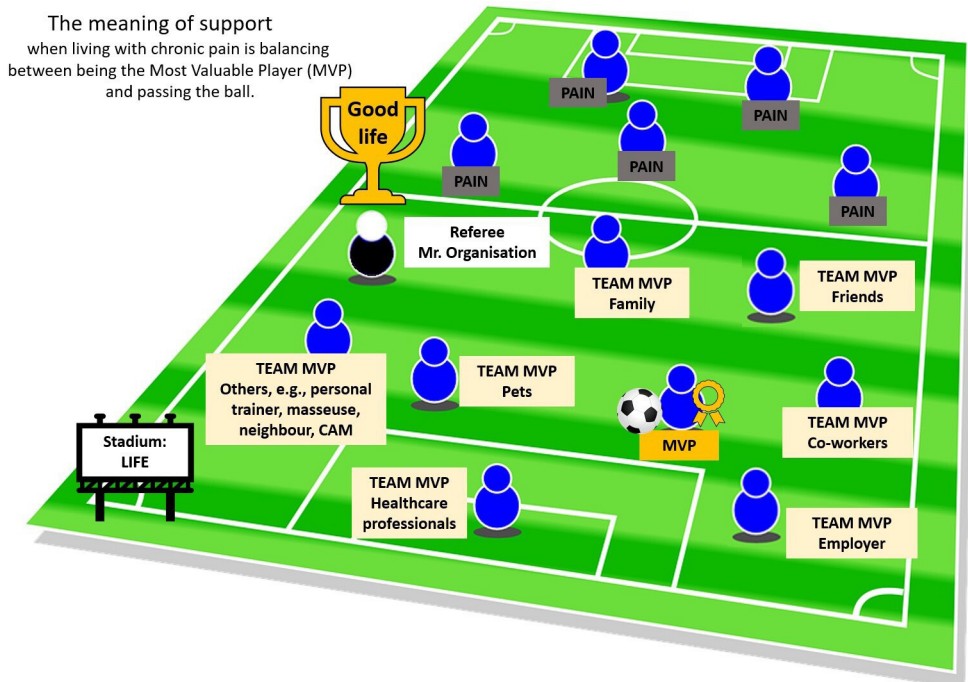

**Figure 1** The match against pain. MVP (person with chronic pain) with the ball (pain management), the team (examples of teammates), the opponent (pain), the referee (representing fair conditions), the stadium (life) and the trophy (symbolising the aspiration for a good life despite living with chronic pain). CAM, complementary and alternative medicine.

about living a meaningful life and being capable despite the pain.

When social networks and healthcare professionals do not listen to persons with chronic pain, believe them, see their capabilities or claim they can no longer be helped, they go to the opposing team. The person with chronic pain might also end up on the opposing team by, for instance, engaging in negative thoughts and behaviours. The field must be just and equitable to ensure the teammates and the MVP are on the same team fighting against the same opponent. For example, healthcare professionals must have fair conditions to stay on the right team and support the MVP instead of considering them a benchwarmer.

The composition of the MVP's team (social networks and healthcare professionals) and the optimal balance between being the MVP and passing the ball (being capable and accepting support to be capable) varies from person to person and over time. Persons with chronic pain feel valued and supported when the healthcare sector and social networks strengthen their capability by encouraging them to be the MVP and pass the ball. The balancing act between the two is the meaning of lived experiences of support.

## DISCUSSION

This study applied an interpretative approach in explaining the meaning of support from the healthcare sector and social networks of persons with chronic pain. The findings indicate that, regardless of who is providing the support, the meaning of support when living with chronic pain is to strengthen the individuals' capability and, when the abilities do not seem enough, feel that someone is fighting together with the individual to regain their capability. Previous studies have mainly performed descriptive analyses, focusing on pain management rather than on the meaning of support.[13–15] Holtrop et al[14] found three primary purposes of important relationships in pain management: providing instrumental support, offering inspiration and motivation, and assisting in decision-making. Our results are similar but show that support strengthens the participants' capability. In line with the present study Holtrop et al[14] found that persons with chronic pain wanted to be recognised as persons rather than their condition and that their lives should be seen as no different from others. Meanwhile, they wanted their limitations due to pain to be accepted. Our study also shows that pets could provide this support, which aligns with Bair et al,[13] who demonstrated that pets can be powerful motivators in pain management. The Bair et al[13] participants relied on support from care managers. Similarly, our results show that support means having someone fight for individuals with chronic pain when their abilities are inadequate.

Acknowledging the patient as an expert and capable person is fundamental to person-centred care.[30] The present study clarifies that persons with chronic pain want to be active and recognised as capable, productive partners in care. Accepting support can strengthen their capability, which is also emphasised in person-centred care.[30] The capabilities approach focuses on human development and social justice, recognising that people's

capabilities are shaped and formed by environmental and social circumstances.[31] A systematic review investigating empirical evidence underpinning the conceptualisations of person-centred care for serious illness found that person-centred care empowers patients and their families by providing information and including them in decision-making actions on their daily lives and care.[32] Considering these results, person-centred care may be valuable in enhancing capability. It would be worthwhile to explore whether this is the case among persons with chronic pain and their close others.

## Methodological considerations

Lindseth and Norberg suggest that phenomenological hermeneutics seeks not to encapsulate the whole truth but to present meanings of a lived phenomenon vis-à-vis interpreting the narrative text.[29] Findings are valid if they represent meaning derived from narrated experiences and illuminate something we want to understand.[29] One-sided opinions can, however, emerge and conscious validation of the interpretation and analysis become important.[29] The hermeneutic spiral, in which pending between understanding (naïve understanding) and explanation (structural analysis), constitutes a reliable approach to validate the findings.[29] Discussions between the authors ensured the interpretations were plausible while not being the only possible options. Additionally, to ensure that the interpretations were reasonable one author (ML) read all the interviews before participating in the analysis.

Qualitative samples should be large enough to understand the studied phenomenon but small enough not to hinder qualitative analysis.[33] In phenomenology saturation is not used to determine the number of participants[34]; fewer participants are needed if the data are rich.[35] The research group decided the collected data sufficed to answer the research question and was not too extensive to capture the meaning of the phenomenon. Most participants in the current study had post high school education, were female and were born in Sweden, which could affect the universality of the findings. Universality is described as an intersubjective understanding of the meaning of lived experiences, meaning that persons can understand the phenomenon better through the findings, even though their situation might not perfectly align with the findings.[29] Because all participants had lived with pain for many years, they had broad experiences of support and provided rich narratives. Follow-up questions were used to ensure that the interviewer understood them correctly. Measures were taken to make participants comfortable (confirming they knew they could ask questions, take breaks, etc) and to encourage sharing their narratives.[23] However, Bruce *et al* described the chronic 'pain journey to acceptance' and that different coping mechanisms are useful depending on where the person is in the journey,[36] which might also translate to support. Some participants grew up with pain, which might have affected our findings. However, participants narrated their lived experiences of support as adults while comparing them to their experiences as children rather than merging them. The need to use Zoom due to the COVID-19 pandemic might have affected the content of the interviews. Still, video interviews are cost-effective and inclusive.[28]

Metaphors were employed in the interpretation process. Ricoeur contends that the metaphor enriches the meaning of a phenomenon through the creative tension of similarities and differences, creating a new understanding.[37] The metaphors demonstrate how language can extend to its limits and affect how we understand the world.

## CONCLUSIONS

For participants, who lived with chronic pain, support means balancing between being capable (the MVP) and willing to accept support (passing the ball), which aligns with the concept of person-centred care. Our findings may be useful for policy-makers, managers and clinical professionals when planning and performing care for persons with chronic pain. Future research should focus on how the healthcare sector can create support to enable persons with chronic pain to be the MVP while being able to pass the ball to their social networks and the healthcare sector.

**Author affiliations**
[1]Institute of Health and Care Sciences, Sahlgrenska Academy, University of Gothenburg, Goteborg, Sweden
[2]University of Gothenburg Centre for Person-Centred Care (GPCC), Sahlgrenska Academy, University of Gothenburg, Goteborg, Sweden
[3]Region Västra Götaland, Sahlgrenska University Hospital, Department of Forensic Psychiatry, Gothenburg, Sweden
[4]Centre for Ethics, Law and Mental Health (CELAM), University of Gothenburg, Gothenburg, Sweden
[5]Healthcare Sciences and E-Health, Department of Women's and Children's Health, Uppsala University, Uppsala, Sweden
[6]Department of Neurobiology, Care Sciences and Society, Karolinska Institute, Stockholm, Sweden
[7]Department of Health Promoting Science, Sophiahemmet University, Stockholm, Sweden
[8]Department of Medicine, Geriatrics and Emergency Medicine, Sahlgrenska University Hospital/Östra, Region Västra Götaland, Gothenburg, Sweden

**Acknowledgements** The authors gratefully acknowledge the study participants for sharing their narratives. We also appreciate the patient organisations (The Swedish Rheumatism Association, The Swedish Association for Survivors of Accident and Injury, Riksförbundet Ehlers-Danlos Syndrom and The Swedish National Organization for Young Rheumatics) for facilitating the recruitment.

**Contributors** VL, SW, IE and ML planned the study. VL performed the data collection. VL, IE, SW, ML, MS and V-AS contributed to the data analysis and interpretation of the findings. VL drafted the manuscript and ML, MS, SW, V-AS and IE critically reviewed it and contributed to the final version prior to submission. IE is responsible for the overall content as guarantor.

**Funding** This work was supported by FORTE grant number (2019-00718) and the University of Gothenburg Centre for Person-Centred Care (GPCC), grant number N/A.

**Competing interests** None declared.

**Patient and public involvement** Patients and/or the public were involved in the design, or conduct, or reporting, or dissemination plans of this research. Refer to the Methods section for further details.

**Patient consent for publication** Not applicable.

**Ethics approval** Ethical approval for this study was granted by the Swedish Ethical Review Authority (Reg nr 2020-02491). In addition, the study was conducted according to the ethical principles outlined in the Declaration of Helsinki. All participants received written and verbal information about the study. Written and verbal informed consent was obtained from all participants and all data were handled confidentially. Participants' identities were removed during transcription. Audio recordings, transcripts and the keycode were safely stored in different locations to protect the participants' identity and personal information.

**Provenance and peer review** Not commissioned; externally peer reviewed.

**Data availability statement** No data are available.

**ORCID iDs**
Veronica Lilja http://orcid.org/0000-0002-5531-6358
Sara Wallström http://orcid.org/0000-0001-7579-4974
Markus Saarijärvi http://orcid.org/0000-0002-7053-6930
Vivi-Anne Segertoft http://orcid.org/0009-0009-7916-668X

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
