## [Reviewer comments · BMJ Open]

ARTICLE DETAILS

TITLE (PROVISIONAL)	Balancing between being the Most Valuable Player (MVP) and passing the ball: A qualitative study of support when living with chronic pain in Sweden
AUTHORS	Lilja, Veronica; Wallström, Sara; Saarijärvi, Markus; Lundberg, Mari; Segertoft, Vivi-Anne; Ekman, Inger

VERSION 1 – REVIEW

REVIEWER	Carmen Caeiro Polytechnic Institute of Setúbal
REVIEW RETURNED	16-Sep-2023

GENERAL COMMENTS	The study entitled “Balancing between being the Most Valuable Player (MVP) and passing the ball: A qualitative study of support when living with chronic pain” aimed to “elucidate the meaning of lived experiences of support from the social networks and healthcare sector in persons with chronic pain”. I applaud the authors for presenting a well-written and interesting manuscript. In my opinion, there are some issues to consider that would improve it. Introduction The introduction clearly presents the rationale for the study. Line 6 – A statement about “a large European study comprising 16 countries” is supported by a study entitled “Prevalence of Chronic Pain and High-Impact Chronic Pain Among Adults - United States”. I’m afraid that this could be secondary citation. If so, I would recommend the authors citing original sources directly. Line 14 – This paragraph is very important to support the relevance of the study. I recommend the introduction of citations that strongly support the main ideas. Additionally, it is mentioned that “this understanding can help in developing tailored interventions for this patient group” - How? Why? Based on previous studies? Line 20 – In my opinion the study’s aim is clear. I wonder why the authors chose the word “elucidate” rather than, for example, “explore”. Methods The overall design is well described and coherent with the aim presented in the introduction section. Line 8 – Participants and setting The inclusion and exclusion criteria are clearly indicated (Inclusion: “≥18 years of age, living in Sweden, and having chronic musculoskeletal pain (defined as “chronic pain arising from musculoskeletal structures”); Exclusion: “Persons who primarily seemed to struggle with other conditions, such as concurrent cancer diagnosis”). However, it is mentioned that “Participants with vast experience of support and willingness to share were purposefully selected”. Which criteria were used for classifying the experience as
---

	“vast”? In my opinion this should be clarified, and some examples could be provided. Line 11 – I found the inclusion of a patient representative very interesting. This person "contributed with experience of living with chronic pain" and “was also part of the research group”. In my opinion, this could be further explored by further indicating this person’s role/ responsibility/ tasks as a member of the research team. This can also be explored as part of “patient and public involvement” content. Findings Structural Analyses, Line 7 – The authors mentioned that “we have used the metaphor of a football match”. I agree that a metaphor can be a powerful and insightful approach, particularly when trying to capture the meaning of lived experiences. Nevertheless, in the context of a phenomenological study, I would expect that the metaphors emerge from the study’s participants descriptions. It is not clear if this metaphor came from “the participants` words” or from the researchers` interpretation. Additionally, the findings section is mainly focused on the description and interpretation of findings with occasional presentation of participants` quotes. In my opinion the reduced number of quotations to support the researchers` interpretations jeopardizes the credibility of the study. I recommend the introduction of more quotations so the authors can demonstrate that the research findings represent plausible information drawn from the participants` original data and are a credible interpretation of the participants` original views. Discussion Methodological considerations Line 5 – It is mentioned that “most participants in the present study were well-educated, female, and born in Sweden, which could affect the generalizability of the findings”. Why is the generalizability an issue in the context of this study? Considering the research paradigm that informs this study, I would expect to find information highlighting that this study’s findings should be considered in terms of theoretical transferability rather than empirical generalizability. Still, transferability is also mentioned – “Because all participants had lived with pain for many years, they had broad experiences of support and provided rich data, enhancing the study's transferability”. I’m afraid that this does not necessarily ensure the transferability of this study’s findings. Transferability represents “the degree to which the results of qualitative research can be transferred to other contexts or settings with other respondents”. (Korstiens, I., Moser, A. (2018) Series: Practical guidance to qualitative research. Part 4: Trustworthiness and publishing. European Journal of General Practice. 24(1); 120-124) The transferability judgement can be facilitated by a thick description of the context in which the study is carried out. I would recommend the authors to revise this information and consider introducing information about the study’s context. Thanks for the opportunity to review this work.
--	---

REVIEWER	Colleen Johnston-Devin Central Queensland University, School of Nursing, Midwifery and Social Sciences
REVIEW RETURNED	20-Sep-2023

GENERAL COMMENTS	BMJ OPEN Balancing between being the Most Valuable Player (MVP) and passing the ball: A qualitative study of support when living with chronic pain
---

	This study aimed to elucidate the meaning of lived experiences of support from the social networks and healthcare sector in persons with chronic pain. It concluded, that for persons with chronic pain, support means balancing between being capable (the MVP) and willing to accept support (passing the ball), which aligns with person-centred care. In elucidating the meaning of support, you are using the word support. You have not answered your research aim. The phenomenological approach was the correct approach for this research. However, It has not been used throughout the research which has resulted in disjointed results. Please refer to the comments below for improvement of this manuscript. STRENGTHS AND LIMITATIONS Line 23 - Further insights was achieved – Insights is plural. Change to were achieved. Line 25 Metaphors are appropriate with the method used. Remove this bullet point. Line 4. The limitation of more female than male participants is recognised. Considering the authors chose the participants, why was this limitation not addressed? Please add this information to the bullet point. Introduction Line 4 expand the notion that chronic pain is a disease for clarity Line 11. Change the word suffering Page 6 Line 9 Not just physiotherapy. Exercise and movement are components of a healthy lifestyle. Perhaps unpack this notion for clarity. Line 14 – do not start a paragraph with because Line 20 Phenomenology strives to understand the essence of a phenomenon. It is the study of phenomena utilising lived experience of the phenomenon as data. The phenomenon in this research is the experience of receiving support. Please rewrite the aim. Participants and setting Line 21 note tense change mid-sentence. Data is collected, not achieved Please provide further detail regarding why the 10 participants were chosen. What determined ‘sufficiently rich’ material? Why were preunderstandings not examined by the rest of the study team? Bias and preunderstandings affect data analysis, not just interview technique. Data Collection Please provide a reference for the three domains of support. Expand on the notion of pilot interviews being deemed relevant. What was revised and did this have a bearing on the information collected? Why did it take 18 months to conduct 10 interviews? Patient and public involvement Did the patient organisations do more than advertise the study? Ethical considerations Participants identities were removed during transcription. Data analysis Please explain the first sentence. Reference is required for the Hermeneutic circle. Please explain what is meant by ‘common naïve understanding’. Themes emerge within phenomenology and coding is not a phenomenological term. Please provide references to support the analysis described. Please explain further the sentence “In the comprehensive interpretation VL and IE compared the pre-understanding, the naïve understanding, and the structural analyses to the existing literature”. Findings
--	--

	Note: You appear to be confused with using your metaphor and theme development. Theme development in this project is not clear. Perhaps van Manen will be of benefit. van Manen (1990) provides a guide to assist the researcher to develop themes but first gives a description of a theme. He believes a theme is “the sense we are able to make of something” (p. 88) and it describes the essence of the notion we are trying to understand. The explication of themes is the means by which the essential structure or form of lived experience is delivered and provide the framework on which to build the story of the phenomenon so that it makes us think, feel and reflectively recognise the lived experience of the phenomenon (van Manen, 1997). It appears that one participant has had pain for 66 years. Did you exclude congenital disorders? For example, if the 77 year old had chronic pain from the age of 11, the experience of receiving support as a child impacted on the support received as an adult. Is this taken into consideration within the findings? Naïve understanding It is not clear how this finding emerged from the data. Please provide excerpts from the transcripts to provide credibility. Structural analyses Please provide a reference for the notion that living with chronic pain is a battle. There is no evidence to support the football match metaphor. Please provide an explanation how this was arrived at. Being the MVP The metaphors have no background to support them. Please rewrite this section using quotes from the transcripts so that the reader can follow an audit trail. Why is dribbling and not being a bench warmer important? More detail about football may strengthen your argument. Being a valuable player and not just the injured one Pets, particularly dogs can sense pain and provide comfort. They are not indifferent. “As a valuable football player, the injury will not matter, as everyone appreciates the player’s efforts on the field and knows their potential”. A player with a broken leg (for example) is never going to be valued on the field. They cannot contribute to running or kicking. While I appreciate you trying to use a metaphor, please reconsider some of your statements. “The teammates show that the player and their abilities are trusted by passing them the ball and allowing them to dribble”. This reads as though the teammates are in charge as they are ‘allowing’. This is metaphorically more like the medical model of care where the physician is in charge and allows the patient to act. I am not sure of the relationship to complimentary therapies in this paragraph. The quote used does not demonstrate support within complementary care or belief adequately. "To receive understanding for my experience of pain, that every individual experiences pain differently, and that it [the pain] is taken seriously is a crucial factor to feel that you're getting support." Why is understanding considered a crucial factor in support? Was this notion explored? Being part of a team. Sentences in this paragraph are not grammatically correct. Please revise. Passing the ball and Being able to pass the ball when you have to These headings are too similar and the differences between them are difficult to discern Needing a substitute without being a benchwarmer Sharing information does not relate to being a burden. Being a burden relates more to asking for help constantly. How many healthcare professionals are writing certificates in their lunch break?
--	---

	The link between the information provided and the heading is not clear. Why is a benchwarmer bad? What is the difference between warming the bench and being a substitute? Interpretation of the whole You have not factored the value of acceptance of pain and the acceptance of the person living well despite limitations that pain might bring. Competing against pain is not a therapeutic strategy. The statement that persons with chronic pain aspire to be important is incorrect and should be removed. This section detracts from the manuscript confusing it further. The metaphor used and figure does not acknowledge that some people are already living well with their chronic pain condition. Passing the ball (pain management) does not account for self-management adequately. Discussion Line 8. The data provided in the findings does not support the statement that support strengthens the participants' capability. Line 12 This was not established in the findings which stated that pets are indifferent The paragraph about complementary therapies is difficult to understand. What therapies were discussed by the participants, and did they feel supported by the practitioners? You have not demonstrated that people living with chronic pain strengthen their capabilities when they accept support. More information about the capabilities approach may help to strengthen this argument. The systematic review mentioned is in the context of serious illness. Please use a reference for patient-centered care and chronic pain. Methodological considerations You have used a reference about Naturalistic Inquiry, not phenomenology. The methodology, theoretical framework and methods used to address the research question must be coherent. Research rigour is achieved when transparency, reflexivity and positioning are made explicit. This has not been achieved in this manuscript.
--	--

VERSION 1 – AUTHOR RESPONSE

Reviewer 1				
I applaud the authors for presenting a well-written and interesting manuscript. In my opinion, there are some issues to consider that would improve it.	We thank reviewer 1 for these kind words and will indeed carefully consider advice and suggestions.			
Introduction				
Line 6 – A statement about “a large European study comprising 16	We are grateful that reviewer 1 noticed this citation error and we have now updated the text and reference list with the correct article.		3. Breivik H, Collett B, Ventafridda V, Cohen R, Gallacher D.	5/7 And 30/8-10

countries” is supported by a study entitled “Prevalence of Chronic Pain and High-Impact Chronic Pain Among Adults - United States”. I’m afraid that this could be secondary citation. If so, I would recommend the authors citing original sources directly.			Survey of chronic pain in Europe: prevalence, impact on daily life, and treatment. European journal of pain (London, England). 2006;10(4):287-333.	
Line 14 – This paragraph is very important to support the relevance of the study. I recommend the introduction of citations that strongly support the main ideas. Additionally, it is mentioned that “this understanding can help in developing tailored interventions for this patient group” - How? Why? Based on previous studies?	We have rephrased a paragraph to make the references support our aim in a clearer way. That understanding could help in developing tailored interventions was a result of our own reflection, and we have now removed this sentence as it created confusion and did not add value to the introduction.	Collaborative relationships with healthcare professionals constitute a support that seems to facilitate self-management of pain (12). Chronic pain is complex and the biopsychosocial model, involving biological, psychological, and social factors, has been successfully used in pain management (20). Evidence-based practice is based on a multimodal approach, including a healthy lifestyle, physiotherapy, and pharmacological and psychological treatment (21).	Collaborative relationships with health care professionals constitute support that facilitates self-management of pain (12). The biopsychosocial model and the multimodal approach have been shown to improve pain management (20, 21). However, it has also been reported that persons with chronic pain feel that healthcare professionals rarely take their condition seriously and that they desire better support from health professionals (3, 15, 22). Due to conflicting evidence and the	6 /4-14

		There is also some evidence for the benefits of complementary therapies (21). Moreover, research has shown that persons with chronic pain desire better support from the healthcare sector (22) and feel that healthcare professionals rarely take their condition seriously (3, 15, 22). Because of conflicting evidence and the complexity of support for persons with chronic pain, there is a need to understand the meaning of support, both within and outside the healthcare system. A deeper understanding of the phenomenon would facilitate the comprehension of the need for support and could aid in shedding light on what kind of support persons with chronic pain want and need. In addition, this understanding can help in	complexity of support for persons with chronic pain, there is a need to understand the meaning of support, both within and outside the healthcare system. A deeper understanding of the phenomenon would facilitate the comprehension of the need for support and could aid in bringing clarity on what kind of support persons with chronic pain want and need.	
--	--	--	---	--

		developing tailored interventions for this patient group.		
Line 20 – In my opinion the study’s aim is clear. I wonder why the authors chose the word “elucidate” rather than, for example, “explore”.	We appreciate this interesting question about elucidating vs exploring. Elucidate is more commonly associated with the method phenomenological hermeneutics. Lindseth and Norberg write in their article from 2022: “The phenomenological-hermeneutical method is a way of elucidating the meaning of life world phenomena developing in time, –in everyday life, in work, in cooperation with others, in research.” We understand the difference as the verb “elucidate” suggests that there is an aspiration to shed light on the phenomenon to facilitate others’ understanding of the meaning of the phenomenon. In contrast, the verb “explore” does not necessarily suggest this aspiration.	Therefore, this study aims to elucidate the meaning of lived experience of support from social networks and the healthcare sector in persons with chronic pain.	No changes made.	
Methods				
Line 8 – Participants and setting The inclusion and exclusion criteria are clearly indicated (Inclusion: “≥18 years of age, living in Sweden, and having chronic musculoskeletal	Some participants wrote down part of their experiences when they responded to the invitation to take part in the study. This was interpreted as a willingness to share. In that same e-mail some participants also expressed that they had experiences of support from both the health care sector and social networks. In contrast, others	Participants with vast experience of support and willingness to share were purposefully selected, as this allows for rich data to be achieved (26)	Five participants mentioned receiving support from the healthcare sector and social networks by starting to share their narratives in the e-mail expressing interest in participating.	7/17-21

pain (defined as "chronic pain arising from musculoskeletal structures"; Exclusion: "Persons who primarily seemed to struggle with other conditions, such as concurrent cancer diagnosis"). However, it is mentioned that "Participants with vast experience of support and willingness to share were purposefully selected". Which criteria were used for classifying the experience as "vast"? In my opinion this should be clarified, and some examples could be provided.	mainly expressed their narrative in relation to one of those. We have rephrased this section to clarify this.		They were purposefully selected as they were willing to share their vast experience of the phenomenon under study, allowing the collection of rich data (26).	
Line 11 – I found the inclusion of a patient representative very interesting. This person "contributed with experience of living with chronic pain" and "was also part of the research group". In my opinion, this could be further explored by further indicating this	We are pleased about this comment as the authors are very excited about our collaboration and its results. We have added a sentence under "patient and public involvement" to emphasise the active participation of VS.	One of the co-authors (VS) is a patient representative.	One of the co-authors (VS) is a patient representative. VS actively participated in data analysis and manuscript preparation.	9/11-12

person`s role/ responsibility/ tasks as a member of the research team. This can also be explored as part of “patient and public involvement” content.				
Findings				
Structural Analyses, Line 7 – The authors mentioned that “we have used the metaphor of a football match”. I agree that a metaphor can be a powerful and insightful approach, particularly when trying to capture the meaning of lived experiences. Nevertheless, in the context of a phenomenologic al study, I would expect that the metaphors emerge from the study`s participants descriptions. It is not clear if this metaphor came from “the participants` words” or from the researchers` interpretation.	The football metaphor was interpreted but inspired by the participants` narratives. We chose to include other quotes as we believed they better demonstrate the meaning in fewer words, but we have now included the football-related quotes again to make the link more explicit.		“They used to ask me,’ Why are you limping?’ I said,’ I played football last weekend,’ and I have never played football. So he [participant’s partner] told me, ‘Now you’re going to stop to tell them that you’ve played football.’ After that, I felt fine. There was no problem. I even have a colleague who says, ‘You’re limping.’ Maybe you should sit down.’ I don’t have to be perfect all the time.” - Participant 4. “You are sort of like a ball being kicked around within the system. And even if you try to tell them ‘No, this is the field, this is the	17/4-9 22/22-23 And 23/1-5

			playing field', it becomes a bit like whatever. [...] But I have been so lucky that these people are helping me out of pure, goodhearted will, even though they do not have time for it. Because their bosses tell them, 'You should only do this or that because these are our resources.' And I mean, they even helped me during lunch and such things. I must be careful when asking them for help because they are so kind-hearted." - Participant 9.	
Additionally, the findings section is mainly focused on the description and interpretation of findings with occasional presentation of participants' quotes. In my opinion the reduced number of quotations to support the researchers' interpretations jeopardizes the credibility of the study. I recommend the introduction of	Lindseth and Norberg (2004, 2019) do not address quotes as part of presenting the research. The method is interpretative, and the findings are mostly validated through the hermeneutic spiral. We did struggle with the quotes interfering with the readability. However, we want the readers to be able to form an opinion of the work, and we have thus chosen to include more quotes, as suggested, and replace some of the previously used ones. We hope this facilitates readability.	Being a valuable player and not just the injured one: "It is about being an important part of society, to contribute... instead of being the one others should take care of. Self-help devices help to achieve that." - Participant 10.	Being a valuable player and not just the injured one: "They used to ask me,' Why are you limping?' I said,' I played football last weekend,' and I have never played football. So he [participant's partner] told me, 'Now you're going to stop to tell them that you've played football.' After that, I felt fine. There was no problem. I even have a	17/4-9

more quotations so the authors can demonstrate that the research findings represent plausible information drawn from the participants' original data and are a credible interpretation of the participants' original views.		Being trusted to dribble: "To receive understanding for my experience of pain, that every individual experiences pain differently, and that it [the pain] is taken seriously is a crucial factor to feel that you're getting support." - Participant 1.	colleague who says, 'You're limping.' Maybe you should sit down.' I don't have to be perfect all the time." - Participant 4. "It is about being an important part of society, to contribute... instead of being the one others should take care of. Self-help devices help to achieve that." - Participant 10. "I find good support in supporting others. You get a reflection of yourself that way. So, maybe that is my best support, to support others. [...] It was perfect to have somebody who needed me." - Participant 3. Being trusted to dribble: "I have to put on a mask in front of people and pretend to be happy. But my	17/16-18 18/1-4 18/20-23 19/5-7
--	--	---	--	---

		Being part of a team: "We [participant and health care team] will do what we can to solve what we can together. I also have to do my part, but we converge in that understanding that there is a problem, and we will try to solve it together—teamwork." - Participant 8.	friend says, "I don't mind that you're low and in pain. You don't need to be happy and energised. We can still have coffee, like getting rid of a 20 kg backpack. He understands." - Participant 7. "My massage therapist sees me. I think that is a need that everyone has, to be seen and listened to." - Participant 2. Being part of a team: "My personal trainer said, 'I have a plan. Let's focus on this so you can continue fighting.' Thanks to her, I am stronger and I feel like things are moving forward." - Participant 1. "I want to take responsibility for my pain and situation myself, but to be able to have someone to	20/5-7 20/9-16
--	--	--	--	------------------------------

		Having teammates when you have been tackled: "When I get this lumbago, I believe it will never pass, and I have to live my whole life like this. And then I talk to him [partner], and he reminds me it will pass and be all right. It is good to be reminded. Otherwise, I'll go into that tunnel, thinking everything will go bad." - Participant 4.	ask for help when I can't bear to deal with it. Like a backup, a support that stands on the side but does not overtake the main responsibility. It is very easy to end up in a subordinate position when you have a chronic condition, that you're in the hands of others. I want to be in charge but still have that support system around me as a backup for when I'm worse. When I do not have that, I have to push myself beyond my limits, and my health deteriorates." - Participant 8.	21/9-11 21/20-23 And 22/1
		Worrying about being a benchwarmer: "But some [healthcare professionals] set aside that time	Having teammates when you have been tackled: "I was advised to change my diet. It helped me so much. It somewhat improved my state because I could affect my situation a little bit myself."	22/8-12

		and try their best within the system they've been put in. It comes with a price for them, but it is significant that they give me that reception and set aside time to listen." - Participant 5.	- Participant 5. "When I get this lumbago, I believe it will never pass, and I have to live my whole life like this. And then I talk to him [participant's partner], and he reminds me it will pass, and I'll be alright. It is good to be reminded. Otherwise, I'll go into that tunnel, thinking everything will go bad." - Participant 4. Worrying about being a benchwarmer: "I don't know how much is okay to... share. What is oversharing? And how much can I share so it becomes enough, so they understand, but it does not become too much? It is a balancing act, and I find it difficult to know where the boundaries are. I dare not take that leap to tell them about my	22/21-23 and 23/1-4
--	--	--	--	------------------------------------

			situation.” - Participant 6. “You are sort of like a ball being kicked around within the system. And even if you try to tell them ‘No, this is the field, this is the playing field’, it becomes a bit like whatever. [...] But I have been so lucky that these people are helping me out of pure, goodhearted will, even though they do not have time for it. Because their bosses tell them, ‘You should only do this or that because these are our resources.’ And I mean, they even helped me during lunch and such things. I must be careful when asking them for help because they are so kind-hearted.” - Participant 9.	
Discussion – Methodological considerations				
Line 5 – It is mentioned that “most participants in the present study	We are very thankful for this comment, as we are delighted to be able to address this incorrect use of	Most participants in the present study were well-educated, female, and born in	Most participants in the current study had post-high-school education, were	27/3-5

were well-educated, female, and born in Sweden, which could affect the generalizability of the findings". Why is the generalizability an issue in the context of this study? Considering the research paradigm that informs this study, I would expect to find information highlighting that this study's findings should be considered in terms of theoretical transferability rather than empirical generalizability.	words.	Sweden, which could affect the generalizability of the findings.	female, and were born in Sweden, which could affect the universality of the findings.	
Still, transferability is also mentioned – "Because all participants had lived with pain for many years, they had broad experiences of support and provided rich data, enhancing the study's transferability". I'm afraid that this does not necessarily ensure the transferability of this study's findings. Transferability	Thanks to this comment and one comment from the second reviewer, we have revised large parts of the text under "methodological considerations" and replaced the references, hopefully improving the section.	Because all participants had lived with pain for many years, they had broad experiences of support and provided rich data, enhancing the study's transferability.	Because all participants had lived with pain for many years, they had broad experiences of support and provided rich narratives. Please also see the other revised parts of "methodological considerations".	27/8-9 26/10-23 And 27/1-23

represents “the degree to which the results of qualitative research can be transferred to other contexts or settings with other respondents”. (Korstiens, I., Moser, A. (2018) Series: Practical guidance to qualitative research. Part 4: Trustworthiness and publishing. European Journal of General Practice. 24(1); 120-124) The transferability judgement can be facilitated by a thick description of the context in which the study is carried out. I would recommend the authors to revise this information and consider introducing information about the study’s context.				
Reviewer 2				
Strengths and limitations				
Line 23 - Further insights was achieved – Insights is plural. Change to were achieved.	We thank reviewer 2 for detecting this subject-verb disagreement and have now changed to the plural form of the verb.	Further insights was achieved with a patient representative who actively participated in the analysis and manuscript	- Further insights were achieved with a patient representative actively involved in the analysis and manuscript	3/24-25

		process.	preparation.	
Line 25 Metaphors are appropriate with the method used. Remove this bullet point.	We are not quite sure why the metaphor could not be considered a strength because it is appropriate with the method, but we have now removed the bullet point as suggested.	- Using a metaphor to describe the findings created a new understanding of the meaning of support when living with chronic pain.	Text removed.	
Line 4. The limitation of more female than male participants is recognised. Considering the authors chose the participants, why was this limitation not addressed? Please add this information to the bullet point.	We have added information as suggested.	- A limitation is that most participants were well-educated, female, and born in Sweden.	- A limitation is that most participants had post-high school education, were female (all eligible participants with other genders were included) and were born in Sweden.	4/3-4
Introduction				
Line 4 expand the notion that chronic pain is a disease for clarity	We considered the comment and changed the wording to clarify that chronic pain is a disease, not just a symptom.	(1). Chronic pain persists or recurs for over 3 months (2) and is viewed as a disease, not just a symptom.	Chronic pain persists or recurs for over 3 months (2) and is classified as a disease on its own, not just a symptom.	5/3-5
Line 11. Change the word suffering	We chose to use this word as it is used in the cited article, but we have reconsidered the wording after this comment.	Suffering from pain is often perceived as invisible to others, which can contribute to feeling unjustly treated in society (8)	Pain is often perceived as invisible to others, which can contribute to feeling unjustly treated in society (8).	5/11-12
Page 6 Line 9 Not just	We agree with this comment and did certainly not intend to	Collaborative relationships with	Collaborative relationships with	6 /4-14

physiotherapy. Exercise and movement are components of a healthy lifestyle. Perhaps unpack this notion for clarity.	state that physiotherapy was the only evidence-based practice. We have, due to reviewer 1's comment already rephrased this sentence and hope that it is now more appropriate.	healthcare professionals constitute a support that seems to facilitate self-management of pain (12). Chronic pain is complex and the biopsychosocial model, involving biological, psychological, and social factors, has been successfully used in pain management (20). Evidence-based practice is based on a multimodal approach, including a healthy lifestyle, physiotherapy, and pharmacological and psychological treatment (21). There is also some evidence for the benefits of complementary therapies (21). Moreover, research has shown that persons with chronic pain desire better support from the healthcare sector (22) and feel that healthcare professionals rarely take their condition seriously (3, 15, 22).	health care professionals constitute support that facilitates self-management of pain (12). The biopsychosocial model and the multimodal approach have been shown to improve pain management (20, 21). However, it has also been reported that persons with chronic pain feel that healthcare professionals rarely take their condition seriously and that they desire better support from health professionals (3, 15, 22). Due to conflicting evidence and the complexity of support for persons with chronic pain, there is a need to understand the meaning of support, both within and outside the healthcare system. A deeper understanding of the phenomenon would facilitate the comprehension of the need for support and could aid in bringing	
---	--	--	--	--

			clarity on what kind of support persons with chronic pain want and need.	
Line 14 – do not start a paragraph with because	We thank reviewer 2 for enlightening us about this. According to our proofreader “starting a paragraph with because is fine as long as you are writing complete sentences.” However, we have replaced the word to not distract readers with disagreements over grammatical issues.	Because of conflicting evidence and the complexity of support for persons with chronic pain, there is a need to understand the meaning of support, both within and outside the healthcare system.	Due to conflicting evidence and the complexity of support for persons with chronic pain, there is a need to understand the meaning of support, both within and outside the healthcare system.	6/10-12
Line 20 Phenomenology strives to understand the essence of a phenomenon. It is the study of phenomena utilising lived experience of the phenomenon as data. The phenomenon in this research is the experience of receiving support. Please rewrite the aim.	For clarity, we have performed phenomenological hermeneutics according to Lindseth and Norberg (2022). Lived experiences constitute a central concept in this method. They also state: “When performing a phenomenological hermeneutical interpretation, our aim is to disclose truths about the essential meaning of being in the life world.” There might be a slight difference compared to “pure” phenomenology. We hope we have understood this comment correctly that the suggestion is to add the word “receiving” to the aim. According to our data, support does not necessarily mean “receiving” support, you can also find support in providing support for someone else, and that is why we chose to avoid including a verb to the aim. Also, we found that a vital	Therefore, this study aims to elucidate the meaning of lived experience of support from social networks and the healthcare sector in persons with chronic pain.	No changes made.	6/15-16

	aspect of support for the participants was to be active and independent. We think using the verb “receiving” would diminish their experiences of support and have thus chosen to use the wording “accepting support” throughout the paper as this indicates a more active role than receiving.			
Participants and setting				
Line 21 note tense change mid-sentence. Data is collected, not achieved	We have changed the wording as suggested.	Participants with vast experience of support and willingness to share were purposefully selected, as this allows for rich data to be achieved	They were purposefully selected as they were willing to share their vast experience of the phenomenon under study, allowing the collection of rich data (26).	7/19-21
Please provide further detail regarding why the 10 participants were chosen.	We hope the changes have clarified the sampling process.	Participants with vast experience of support and willingness to share were purposefully selected, as this allows for rich data to be achieved (26). As maximum variation sampling allows uncovering common meaning across demographic differences (26), a diversity of experiences of support, age, geographic location, sick leave rate, and	Five participants mentioned receiving support from the healthcare sector and social networks by starting to share their narratives in the e-mail expressing interest in participating. They were purposefully selected as they were willing to share their vast experience of the phenomenon under study, allowing the collection of rich data (26).	7/17-23 And 8/1-5

		background diagnosis was strived for even if most potential participants were women. Eight participants were initially included, and their narratives were deemed sufficient to answer the study's research question. However, another two participants were interviewed to achieve greater variation in educational level. None of the participants declined to participate.	Maximum variation sampling allowed the discovery of common meanings across demographic differences (26). Therefore, a diversity of experiences of support, age, geographic location, sick leave rate, and background diagnosis was sought even though most potential participants were women. Eight participants were initially included, and their narratives were deemed sufficient to answer the study's research question. Another two participants were interviewed to achieve greater variation in education level. None of the participants declined to participate.	
What determined 'sufficiently rich' material?	We have explained this under "methodological considerations" ("The research group decided that the collected data sufficed to answer the research question and were not too extensive to capture the meaning of the phenomenon.") But have now added a clarification in the "participants and setting"-section.	The material was then deemed sufficiently rich, and after discussions within the research group, inclusion was halted.	The material was considered rich enough to find meanings of support. After discussions in the research group, inclusion was halted at 10 participants.	8/5-6

Why were pre-understandings not examined by the rest of the study team? Bias and Pre-understandings affect data analysis, not just interview technique.	This was a noteworthy perspective that we enjoyed discussing. Pre-understanding within phenomenological hermeneutics (as we interpret Lindseth and Norberg's articles) is not about bias but is seen as necessary to make interpretations. We constantly use and revise our pre-understanding as we learn more about a phenomenon. Our reasoning for including the pre-understanding as supplementary material is to be determine the progress from the first pre-understanding to the final findings in the manuscript, and also to be transparent. Because VL is new to the field and the main contributor to the manuscript, we thought it would be most interesting to follow the progress of her pre-understanding. However, we will consider writing down the pre-understanding of all the authors if we were to do a similar study in the future.		No changes made.	
Data collection				
Please provide a reference for the three domains of support.	The importance of the domains is described and referenced in the introduction. We have added a sentence to clarify this.	The guide covered three domains of support: the healthcare sector, social networks, and how support from social networks could be integrated within care.	The guide included three domains of support: the healthcare sector, social networks, and how support from social networks could be integrated into care. The three domains were chosen based on their previously described importance (3, 12-16, 19-22).	8/16-19

Expand on the notion of pilot interviews being deemed relevant.	We have now clarified this notion of pilot interviews and how they are relevant.	The narratives from the two pilot interviews were deemed relevant and thus included in the data analysis.	The narratives from the two pilot interviews were deemed relevant, as they answered the research question and were thus included in the data analysis.	9/1-3
What was revised and did this have a bearing on the information collected?	We have now clarified this point.	The interview guide was piloted in the first two interviews and slightly revised.	The interview guide contained open-ended questions with suggestions for additional probing questions. It was piloted in the first two interviews and revised by changing from the question “Which persons outside of the healthcare sector have you gotten support/help from to “Which persons outside of the healthcare sector have been important to you regarding your pain?” The final version of the question better facilitated narratives about social networks.	8/19-23 And 9/1
Why did it take 18 months to conduct 10 interviews?	The interviews took 18 months because the first author had other professional commitments and was not working full-time. Because we do not wish to exceed the word limit further, we chose not to add information about this in the manuscript.		No changes made.	
Patient and public				

involvement				
Did the patient organisations do more than advertise the study?	The patient organisations assisted with recruitment by posting it on their Facebook page or sending their members an e-mail (as described under participants and setting). Still, they can spread the results if they wish. We have added this point to the manuscript.	Four patient organizations contributed to recruiting study participants.	Four patient organisations contributed to recruiting study participants and will contribute to disseminating the findings.	9/10-11
Ethical considerations				
Participants identities were removed during transcription.	We have used the suggested wording of this comment.	Participants' identities were removed during data transcripts.	Participants' identities were removed during transcription.	9/19-20
Data analysis				
Please explain the first sentence. Reference is required for the Hermeneutic circle.	We have added the reference for the hermeneutic spiral. The first sentence, “The data were analysed with phenomenological hermeneutics”, is explained by the following sentences and paragraphs within the same heading. However, since this seems unclear, we have put headings for the naïve understanding, structural analyses and interpretation of the whole. We hope this revision adds clarity. Additionally, we reworded the first sentence about the interpretation of the whole, so consistent language appears throughout the manuscript to avoid further confusion.	The data were analysed with phenomenologica l hermeneutics. The method, influenced by Ricoeur's theory of interpretation and developed by Lindseth and Norberg (23), involves three intertwined phases: naïve understanding, structural analysis, and comprehensive understanding. Through the hermeneutic spiral, the phases are constantly overlapping, revisited and compared to each other to move between explanation and understanding by	Data were analysed with phenomenologica l hermeneutics. The method, influenced by Ricoeur's theory of interpretation and developed by Lindseth and Norberg (23), involves three intertwined phases: naïve understanding, structural analysis, and comprehensive understanding. Through the hermeneutic spiral, the phases are constantly overlapping, revisited and compared to each other to move between explanation and understanding by	10/2-7 11/7-8

		interpretation of the whole and the parts. In the comprehensive interpretation VL and IE compared the pre-understanding, the naïve understanding, and the structural analyses to the existing literature.	interpretation of the whole and the parts (30). In interpreting the whole VL and IE compared the pre-understanding, the naïve understanding, and the structural analyses several times to identify inconsistencies. Please see the headings under “data analysis.”	10/9 And 10/13 And 11/6
Please explain what is meant by ‘common naïve understanding’.	We have revised this sentence to improve its clarity.	Each interview was read several times. VL then formulated a naïve understanding for each interview before cultivating a common naïve understanding.	Each interview was read several times. VL formulated a naïve understanding for each interview before constructing a merged naïve understanding of all interviews.	10/10-11
Themes emerge within phenomenology and coding is not a phenomenological term. Please provide references to support the analysis described.	We thank reviewer 2 for pointing out our incorrect word use concerning the chosen method. For clarity, we would like to point out that the method is not phenomenology but phenomenological hermeneutics. Lindseth and Norberg, the primary methodological reference within the manuscript, state that the structural analysis can be conducted in several	The text of each interview was divided into meaning units and then coded. All text was considered, but only text associated with the study's aim was included in the structural analysis. The codes were	The text of each interview was divided into meaning units and condensed. All text was considered, but only text associated with the study's aim was included in the structural analyses. The condensed	10/15-21 And 11/1-2 And 11/4 (table heading)

	ways. However, we would want to use the more familiar word. We have thus replaced “codes” with “condensed meaning units”, as the meaning units were condensed into one or a few words.	continuously compared to the naïve understanding. The interviews were read through again, and the naïve understanding was revised and compared to the structural analysis. This process was repeated several times. Eventually, tentative themes and sub-themes were formulated and compared to the codes and the naïve understanding of the interviews. VL and IE constantly discussed and reformulated the tentative findings before consulting the other authors.	meaning units were continuously compared to the naïve understanding. The interviews were read through again, and the naïve understanding was revised and compared to the structural analyses. This process was repeated several times. Eventually, tentative themes and sub-themes were formulated and compared to the condensed meaning units and the naïve understanding. VL and IE continuously discussed and reformulated the tentative findings before consulting the other authors.	
Please explain further the sentence “In the comprehensive interpretation VL and IE compared the preunderstanding , the naïve understanding, and the structural analyses to the existing literature”.	We have provided a more detailed explanation, as suggested.	In the comprehensive interpretation VL and IE compared the pre-understanding, the naïve understanding, and the structural analyses to the existing literature. The comprehensive understanding was then discussed between all	In interpreting the whole VL and IE compared the pre-understanding, the naïve understanding, and the structural analyses several times to identify inconsistencies. The analysis was revised until all parts were consistent. The underlying meaning in the	11/7-8 And 12/1-11

		authors. MS, SW, and VS read and gave feedback on the findings. ML read all the interviews and the results to ensure the interpretations were reasonable.	data was reflected on and compared to the existing literature, such as previous studies, the work of the philosopher Ricoeur, and the underpinnings of person-centred care, yielding a new understanding. ML read all the interviews and the findings to ensure the interpretations were reasonable before giving feedback on the naïve understanding, structural analyses, and interpretation of the whole. The understanding of the meaning of the findings was discussed among all authors several times to connect their perspectives, knowledge, and understandings. The interpreted metaphor was developed through discussions among all authors based on the link to the naïve understanding and structural analyses. When consensus on the meaning of the findings and the metaphor was	
--	--	--	--	--

			reached, the interpretation of the whole was formulated.	
Findings				
Note: You appear to be confused with using your metaphor and theme development. Theme development in this project is not clear. Perhaps van Manen will be of benefit. van Manen (1990) provides a guide to assist the researcher to develop themes but first gives a description of a theme. He believes a theme is “the sense we are able to make of something” (p. 88) and it describes the essence of the notion we are trying to understand. The explication of themes is the means by which the essential structure or form of lived experience is delivered and provide the framework on	We greatly appreciate the advice and the effort reviewer 2 has put into sharing the reference with us. We are unsure if we understood this comment correctly, but we interpret it as the issue that includes the metaphor in the structural analysis. We have discussed this thoroughly. Lindseth and Norberg (2004, 2022) suggest that the reason for doing phenomenological hermeneutics is to create an understanding of the derived meaning, and (as quoted in a comment response to “being the MVP” further down in this revision), they encourage the use of artistic language (e.g., metaphors). However, they do not state where the metaphors can or should be included in the analysis. They do note that the structural analysis can be performed in several ways and that their example of a thematic analysis is just an example. So, we have now contemplated our options with this manuscript, and it would be possible not to introduce the football metaphor until the interpretation of the whole. However, we believe that raising it already in the structural analysis facilitates the readers’ understanding. By putting each sub-theme, theme, and main theme in relation to the metaphor, we believe it becomes more evident to the reader how the		No changes made/changes already made due to other comments.	

which to build the story of the phenomenon so that it makes us think, feel and reflectively recognise the lived experience of the phenomenon (van Manen, 1997).	metaphor relates to the data and what it means. Thus, we have decided to retain the metaphor within the structural analysis but hope that the changes made - thanks to the other comments of both reviewers - will lead to the metaphor making more sense now.			
It appears that one participant has had pain for 66 years. For example, if the 77 year old had chronic pain from the age of 11, the experience of receiving support as a child impacted on the support received as an adult. Is this taken into consideration within the findings?	We appreciate that the child's perspective on support differs significantly from that of an adult. Our experience is that the participants who had lived with pain since childhood narrated their experience as an adult but could compare it to their experiences as a child to convey their meaning of support better. We have added a sentence about this possibility in the discussion section.		Some participants grew up with pain, which might have affected our findings. However, participants narrated their lived experiences of support as adults while comparing them to their experiences as children rather than merging them.	27/15-17
Did you exclude congenital disorders?	We have clarified this issue and hope it is clearer now.	Persons who primarily seemed to struggle with other conditions, such as concurrent cancer diagnosis, were excluded. Participants who mainly wanted to share their narratives about musculoskeletal pain but had previously undergone cancer treatment	Persons who primarily seemed to struggle with other conditions and congenital diseases, such as concurrent cancer diagnosis, were excluded. Participants who mainly wanted to share their narratives about musculoskeletal pain but had congenital diseases,	7/10-14

		or had another pain-related diagnosis were not excluded.	undergone cancer treatment or had another pain-related diagnosis were not excluded.	
Naïve understanding				
It is not clear how this finding emerged from the data. Please provide excerpts from the transcripts to provide credibility.	We hope the process of the naïve understanding has become clearer now after revising the text with the aspiration to clarify what we mean by “common” naïve understanding under the section “data analysis”. Lindseth and Norberg (2004) state that “The text is read several times to grasp its meaning as a whole. [...] The naïve understanding of the text is formulated in phenomenological language. It is regarded as a first conjecture and it has to be validated or invalidated by the subsequent structural analysis. Thus, the naïve understanding guides the structural analysis”. The naïve understanding is thus validated by the structural analysis, not excerpts from the transcripts. The naïve understanding emerges from reading the interviews several times. It is then revised through the hermeneutic spiral of understanding and explaining while performing the structural analysis and interpretation of the whole. Our experience is that other phenomenological studies		No changes made.	

	(such as Lundin et al., which we have referenced in our manuscript) do not usually provide quotes in this section. We have thus chosen not to include quotes in the naïve analysis.			
Structural analyses				
Please provide a reference for the notion that living with chronic pain is a battle. There is no evidence to support the football match metaphor. Please provide an explanation how this was arrived at.	We greatly appreciate this comment, as our aspiration with the metaphor was to make the findings more understandable and accessible to the readers. Since it was not clear where the metaphor came from, we are pleased to have the opportunity to address this. Firstly, we have removed a sentence under the heading “Balancing between being the Most Valuable Player (MVP) and passing the ball in the match against pain” and instead added a more elaborating paragraph under the heading “structural analysis.” As mentioned at the end of the text, under “Balancing between being the Most Valuable Player (MVP) and passing the ball in the match against pain”, the thought is that the interpreted battle against chronic pain should be illustrated through the themes and sub-themes. The struggle we are referring to can be found described under the following sub-themes (but also in the naïve understanding) of the first submitted manuscript version: Being a valuable player and	In these analyses we have used the metaphor of a football match. The main theme, themes, and sub-themes are described below and in Table 3. The participants sought to be recognized foremost as the persons they were, with unique personalities and experiences. To the participants a diagnosis was important to get their experiences acknowledged, understanding their pain and	Text removed. The main theme, themes, and sub-themes are described below and in Table 3. The metaphor of a football match is used throughout the designation of the main theme, themes and sub-themes to elucidate the meaning of support. The metaphor is related to participants’ narratives and elaborated on under each heading. Participants sought to be recognised as the persons they were, with unique personalities and experiences, which could often be difficult to achieve. To participants a diagnosis was	15/2-5 16/13-14 18/13-16

	not just the injured one: “When perceived as a product of their pain, they felt excluded and viewed as someone who could not accomplish much.” “Participants also wanted healthcare professionals to recognize them as the persons they were.” Being trusted to dribble: “To the participants a diagnosis was important to get their experiences acknowledged, understanding their pain and being believed.” “The participants sometimes felt they received better support within complementary therapies compared to traditional care.” Being part of a team: “The participants often felt alone struggling with pain. Navigating the healthcare system and the social insurance agency's policies was difficult. Fighting different systems alone evoked feelings of being diminished, vulnerable, and powerless” Having teammates when you have been tackled: “Participants felt hopeless when told by health care professionals that there was nothing left to be done after undergoing several medical treatments or when test results failed to reveal the cause of their pain.” Needing a substitute without being a benchwarmer: “The participants considered	being believed. A diagnosis meant validating their condition and was also experienced as facilitating being believed, trusted, and understood by the social networks.	important to have their pain experiences taken seriously, understanding their pain, and being believed, but it was often perceived as a challenging process. A diagnosis meant validating their condition and was also experienced as facilitating being believed, trusted, and understood by the social networks.	
--	---	--	---	--

	it challenging to determine what they could share about their situation with their networks (particularly family and friends) without appearing as a burden. Not wanting to strain their networks or cause worry discouraged the participants from including them in their care. Some healthcare professionals blamed the participants for their situation, whereas others went beyond their duties to support the participants by writing certificates during their free time (e.g., lunch break). [...] At the same time, the feeling of being a burden was enhanced because the healthcare professionals had to sacrifice their spare time to offer that support.” We have added clarifying sentences to the first two sub-themes to ensure the struggle is better understood.			
Being the MVP				
The metaphors have no background to support them. Please rewrite this section using quotes from the transcripts so that the reader can follow an audit trail. Why is dribbling and not being a bench warmer important? More detail about football may	The theme is explained through its sub-themes, but to ensure that it is clear to the reader that the metaphor is an interpretation of the data, we added a few words here. Lindseth and Norberg (2004) write, "Narrative language often involves poetic expressions. Poetic language makes the words mean as much as they can and creates mood, which reveals possible ways of being in the world and ‘shows	In football terms they aspired to be the MVP in all aspects of their lives. The teammates show that the player and their abilities are trusted by passing them the ball and allowing them to dribble..	Contemplating the metaphor, this can be interpreted as they aspired to be the MVP in all aspects of their lives. The teammates show that the player and their abilities are trusted by passing them the ball and encouraging	16/7-8 19/9-11 20/18-

strengthen your argument.	a deeper mode of belonging to reality' (28), while scientific language reduces the polysemy of language (29). Thus sometimes we use poetic expressions, metaphors or sayings in order to convey the interpreted meaning.” We have interpreted this as an encouragement to use the interpreted metaphor, as we believe it further elucidates the meaning and possibly creates a better understanding (which is the aspiration of the method). As suggested, We have added more information about our assumptions regarding football when we tie the metaphor to the sub-themes.	Reducing loneliness by accepting support from others could be analogous to being part of a football team. The individual is stronger with teammates and stands a better chance of beating the other team (i.e., overcoming chronic pain). Regaining hope could be interpreted as having someone to pass the ball to when the football player can no longer dribble on their own (when the pain is unbearable, and they do not know how to move on). By passing the ball, the hope remains that the team can control the ball instead of losing it to the other team.	them to dribble, meaning taking responsibility for the next move. Reducing loneliness by accepting support from others could be analogous to being part of a football team. The individual is stronger with teammates and has a better chance of beating the other team (i.e., overcoming chronic pain). Being part of a team also means being included and not alone. Regaining hope could mean having a teammate take the ball when the football player has been tackled and cannot dribble on their own (when the pain is unbearable, and they do not know how to move on). By having a teammate take the ball (pain management), the hope remains that the team can control the ball instead of losing it to the opposing team (the pain).	21 21/13-19
---	--	---	---------------------------

			There is hope that there is something else to try, even though all options for pain management are exhausted and someone is fighting with you when you feel like your abilities are not enough to fight the pain on your own.	
Being a valuable player and not just the injured one				
Pets, particularly dogs can sense pain and provide comfort. They are not indifferent.	We appreciate this comment. We have made a language-related mistake. What we meant to convey is that the pain does not matter to pets. They still love their owners unconditionally. We have corrected this confusion.	Pets could also contribute to this kind of support, as they were indifferent to the participants' pain and provided unconditional love and companionship regardless.	Pets could also contribute to this support, providing unconditional affection and companionship.	17/11-12
“As a valuable football player, the injury will not matter, as everyone appreciates the player’s efforts on the field and knows their potential”. A player with a broken leg (for example) is never going to be valued on the field. They cannot contribute to running or	This is an intriguing argument to us, as we have never viewed an injured player in this light and were under the impression that injured football players were indeed valued despite their injuries as they receive the best available medical help and play all the important matches as soon, and as much, as possible. We thus thought this was a good metaphor for our data (that the meaning of support when living with chronic pain can be that you are still being valued even though you’re in pain and might not be able to do everything to the same	As a valuable football player, the injury will not matter, as everyone appreciates the player’s efforts on the field and knows their potential.	Valuable football players are still useful to the team when injured, as everyone appreciates their efforts and knows their potential. They are not regarded as “that injured player.” They are still valuable, and everyone is eager to see them return to the field.	18/7-10

kicking. While I appreciate you trying to use a metaphor, please reconsider some of your statements.	extent as you would have without pain). However, this comment made us question our pre-understanding. Of course, many other injuries besides broken legs can happen to football players. Still, we found an article looking into traumatic leg fractures and they concluded that most injured players did not perform less than before their injury but had a significant loss of playing time. (Reference: Lavoie-Gagne O, Gong MF, Patel S, Cohn MR, Korrapati A, Forlenza EM, Barmonyallah M, Parvaresh KC, Wolfson TS, Forsythe B. Traumatic Leg Fractures in UEFA Football Athletes: A Matched-Cohort Analysis of Return to Play, Reinjury, Player Retention, and Performance Outcomes. Orthop J Sports Med. 2021 Sep 8;9(9):23259671211024218. doi: 10.1177/23259671211024218) This makes us still believe that the metaphor is relevant and facilitates understanding the findings. However, we now have realised that this is not everyone's view of injured players. We do not believe citing the article in our results section is appropriate, but we have clarified our assumptions in the text instead.			
"The teammates show that the player and their abilities are trusted by passing them the ball and	We are happy reviewer 2 spotted this, as this is the opposite of what we are trying to convey. We have reconsidered and changed the wording.	The teammates show that the player and their abilities are trusted by passing them the ball and allowing	The teammates show that the player and their abilities are trusted by passing them the ball and	19/9-11

allowing them to dribble". This reads as though the teammates are in charge as they are 'allowing'. This is metaphorically more like the medical model of care where the physician is in charge and allows the patient to act. I am not sure of the relationship to complimentary therapies in this paragraph.	We also divided the paragraph.	them to dribble. The participants sometimes felt they received better support within complementary therapies compared to traditional care. They felt listened to and perceived as capable persons. This complimentary support, however, was not always affordable because of the participants' often strained economy. Being believed and listened to when sharing experiences of pain could be interpreted as a football player being trusted by other teammates. The teammates show that the player and their abilities are trusted by passing them the ball and allowing encouraging them to dribble.	encouraging them to dribble, meaning taking responsibility for the next move. Participants sometimes felt they received better support within complementary therapies compared to traditional care. They felt listened to and perceived as capable. This complimentary support, however, was not always affordable because of the participants' often strained economy. "My massage therapist sees me. I think that is a need that everyone has, to be seen and listened to." - Participant 2. Being believed and listened to when sharing experiences of pain could be viewed as a football player being trusted by other teammates. The teammates show that the player and their abilities are trusted by	19/1-11
--	---------------------------------------	--	--	----------------

			passing them the ball and encouraging them to dribble, meaning taking responsibility for the next move.	
The quote used does not demonstrate support within complementary care or belief adequately. "To receive understanding for my experience of pain, that every individual experiences pain differently, and that it [the pain] is taken seriously is a crucial factor to feel that you're getting support." Why is understanding considered a crucial factor in support? Was this notion explored?	We have now included a quote about complementary therapies. We have clarified the text regarding understanding and replaced the previously used quote.	When the expectations from others did not clash with their abilities, while still being trusted with tasks they could perform, the participants' view of themselves as capable persons was reinforced. "To receive understanding for my experience of pain, that every individual experiences pain differently, and that it [the pain] is taken seriously is a crucial factor to feel that you're getting support. - Participant 1.	"My massage therapist sees me. I think that is a need that everyone has, to be seen and listened to." - Participant 2. Being believed, trusted, and understood by others was a support, and it also encouraged that expectations from others did not clash with the participants' abilities. When participants were trusted with tasks they could perform, their view of themselves as capable persons was reinforced. "I have to put on a mask in front of people and pretend to be happy. But my friend says, "I don't mind that you're low and in pain. You don't need to be happy and energised. We can still have	19/5-7 18/16-19 18/20-23

			coffee, like getting rid of a 20 kg backpack. He understands.” - Participant 7.	
Being part of a team				
Sentences in this paragraph are not grammatically correct. Please revise.	We have revised this paragraph and asked our proofreader to review this section with extra attention.	When they found regular contact with a healthcare professional willing to help and possibly involve other professionals, or when their networks was ready to help them fight the system, they did not feel as lonely. This regular contact enhanced the participants' sense of capability. Having support from others, they felt stronger and could accomplish more.	Participants often felt alone, struggling with pain. Navigating the healthcare system and the social insurance agency's policies was difficult. Fighting different systems alone evoked feelings of being diminished, vulnerable, and powerless. When they found regular contact with a health care professional willing to help and possibly involve other professionals, or when their networks helped them fight the system, they did not feel as lonely. Regular contact enhanced the participants' sense of capability. Having support from others, they felt more confident and could accomplish more.	19/20-22 And 20/1-4
Passing the ball and Being able to pass the ball				

when you have to				
These headings are too similar and the differences between them are difficult to discern	We are thankful for this comment and have now revised the wording of the sub-theme to make a more apparent distinction between the two without losing the meaning.	Being able to pass the ball when you have to	Having teammates when you have been tackled. Please also see related changes throughout the manuscript.	21/1
Needing a substitute without being a benchwarmer				
Sharing information does not relate to being a burden. Being a burden relates more to asking for help constantly. How many healthcare professionals are writing certificates in their lunch break? The link between the information provided and the heading is not clear. Why is a benchwarmer bad? What is the difference between warming the bench and being a substitute?	Our participants' narratives revealed that sharing information, not just asking for help, could mean feeling like a burden and that this was an important aspect of the meaning of support. We've now included a quote so this becomes clear. Many participants stated that health care professionals did work on their lunch break to help them, and putting up the certificates we believed was a concrete example of how. However, since this raises questions, we have made the description more general. We are once again happy to gain a new perspective on our metaphor. We have clarified the concept of being a benchwarmer. Making a substitute means you are part of the starting line-up and get a break. You can play again, maybe not this particular match, but the next.	Some healthcare professionals blamed the participants for their situation, whereas others went beyond their duties to support the participants by writing certificates during their free time (e.g., lunch break). That kind of support made	"I don't know how much is okay to... share. What is oversharing? And how much can I share so it becomes enough, so they understand, but it does not become too much? It is a balancing act, and I find it difficult to know where the boundaries are. I dare not take that leap to tell them about my situation." - Participant 6. Some health care professionals blamed participants for their situation, whereas others went beyond their regular duties to support participants by working during	22/8-12 22/14-16

	However, we now recognise that assuming all readers will know this is not feasible, we reworded the theme's text to avoid confusion.	a huge difference, as the participants needed the certificates to apply for services and sick leave. Fear of being a burden could be seen as fear of losing the title of MVP and becoming a benchwarmer, i.e., slowing down the team. Needing a substitute without being a benchwarmer	their free time (e.g., lunch break). Such support made a huge difference, as participants needed their help. Fear of being a burden due to pain could be seen as fear of losing the title of MVP and becoming a benchwarmer, i.e., always on the bench without the opportunity to participate and contribute. Worrying about being a benchwarmer Please also see related changes to the manuscript.	22/18-20 22/3
Interpretation of the whole				
You have not factored the value of acceptance of pain and the acceptance of the person living well despite limitations that	We value this comment, as it is clear to us that our main points certainly do not come across as intended. Our aim is to elucidate the meaning of lived experiences of support, regardless of what the social networks or contact with healthcare might be to the MVP (i.e. also irrespective of	As the main theme suggests, the meaning of the participants' lived experiences of support from the healthcare sector and networks is to balance being a	As the main theme suggests, the meaning of the participants' lived experiences of support from the healthcare sector and social networks is to balance being a	23/7-23 And 24/1-23

pain might bring. Competing against pain is not a therapeutic strategy. The statement that persons with chronic pain aspire to be important is incorrect and should be removed. This section detracts from the manuscript confusing it further. The metaphor used and figure does not acknowledge that some people are already living well with their chronic pain condition. Passing the ball (pain management) does not account for selfmanagement adequately.	whether the person in mind is already living well with their chronic pain). Through the metaphor, we sought to create a more accessible understanding of the meaning of support when living with chronic pain, not to suggest a therapeutic strategy. We are unsure what part of our findings leads the reader to believe that this is a suggested therapeutic strategy or that self-management (a huge part of being the MVP) is not accounted for. Still, we see this as an urgent priority to address. We acknowledge acceptance in the structural analyses (specifically under the first sub-theme), whereas the next interpretative step (interpretation of the whole) is not as pronounced, just as the other findings under each sub-theme. This is due to the interpretation of the whole being a more interpretative “layer”, which (as we’ve understood Lindseth and Norbergs’ articles) should build upon the naïve understanding and the structural analyses but should also present a more universal meaning (and not just repeat the structural analyses). This quote about the interpretation of the whole from Lindseth and Norberg (2004) might facilitate the understanding of our reasoning: “The focus is not on what the text says but on the possibilities of living in the world that the interview text opens up.” Our claim that	capable person and accepting help from others to continue being that capable person. Using football terms, the MVP and the team compete against pain (see Figure 1). All people want to feel needed and useful. Any football player's dream is to be the MVP whose actions ultimately determine the match's outcome. Persons with chronic pain are no different, i.e., they aspire to be important. However, not even the MVP can win a match without team support. Many football players (especially injured players) will not play during the full 90 minutes but may substitute in the second half of the game without making them any less valuable to the team. They are not seen as a burden and can still be the MVP. Everyone recognizes those football players' previous efforts and potential.	capable person and accepting help from others to continue being that capable person. Being capable means being recognized for who you are as a person and your qualities, to contribute, to be trusted and listened to. Accepting support means to not be alone, having someone fighting with you in order to enhance one’s capability, but also to worry about being a burden. In football terms, being capable corresponds to being the MVP and accepting support corresponds to passing the ball. Developing the interpreted football metaphor; the MVP and the team compete against pain (see Figure 1). Many football players dream of being the MVP, whose actions ultimately determine the match’s outcome. Our interpretation of the findings is that persons with	
---	--	--	--	--

	persons with chronic pain also seek to be important ties into this explanation. It is another step in the interpretation of the findings. We found that this is an essential meaning of being the MVP (a capable person), to be important, and it is derived from the process of reaching this interpreted whole (which is now better described in the method section). We have thus chosen to keep this word instead of disregarding it. We have made changes to the text beneath “interpretation as a whole” with this comment in mind and hope that our main points come across better and that the misunderstandings have now been sorted out.	The teammates will pass them the ball, let the MVP dribble and await the MVP to pass it back. The organisation will provide the best available medical help and tailored training programs to help those players reach their full potential in every match. However, if you do not pass these players the ball or ensure they can pass to you when they are blocked, you've shifted over to the opposite team. When the social networks and healthcare professionals do not listen, believe them, see their capabilities, or claim that the persons with chronic pain can no longer be helped, they step over to the opposite team. The MVP might also end up on the opposing team by for instance engaging in negative thoughts and behaviours. How will the MVP win the match against pain when the teammates or they themselves	chronic pain are no different, i.e., they aspire to be important. However, not even the MVP can win a match without team support. They need the team to believe in their capability and support them to reach their full potential. In accordance with how the MVP takes the lead, we would argue that persons with chronic pain take the lead in their daily lives and care by dribbling the ball (pain management). However, there must be a balance, as no team will win a match through only having the MVP dribbling, i.e., it is also necessary to pass the ball. Passing the ball does not mean losing the title as MVP; instead, it means boosting the chances of winning the match through the help of teammates. Living with chronic pain often leads to accepting a life	
--	--	---	---	--

		become the enemy? The field must be just and equitable to ensure the teammates and the MVP are on the same team fighting against the same opponent. Healthcare professionals must have fair conditions to stay on the right team and support the MVP. As with any successful football team, the team needs its MVP, and the MVP needs the team. The MVP's capability needs to be recognized within a reliable and supportive team for the MVP to perform well. The participants are the most influential persons in their own lives, and they need to be trusted as a valuable team member. People with pain feel valued and supported when the healthcare sector and social networks strengthen their abilities and capability.	with pain. Aspiring to win the match against pain is not about being pain-free but about living a meaningful life and being capable despite the pain. When social networks and healthcare professionals do not listen to persons with chronic pain, believe them, see their capabilities, or claim they can no longer be helped, they go to the opposing team. The person with chronic pain might also end up on the opposing team by, for instance, engaging in negative thoughts and behaviours. The field must be just and equitable to ensure the teammates and the MVP are on the same team fighting against the same opponent. For example, healthcare professionals must have fair conditions to stay on the right team and support the MVP instead of considering them a benchwarmer.	
--	--	---	---	--

			The composition of the MVP's team (social networks and healthcare professionals) and the optimal balance between being the MVP and passing the ball (being capable and accepting support to be capable) varies from person to person and over time. Persons with chronic pain feel valued and supported when the healthcare sector and social networks strengthen their capability by encouraging them to be the MVP and pass the ball. The balancing act between the two is the meaning of lived experiences of support. [INSERT FIGURE 1 HERE] Legend: Figure 1. The match against pain. Most Valuable Player/MVP (person with chronic pain) with the ball (pain management), the team (examples of teammates), the opponent (pain), the referee	
--	--	--	---	--

			(representing fair conditions), the stadium (life), and the trophy (symbolising the aspiration for a good life despite living with chronic pain).	
Discussion				
Line 8. The data provided in the findings does not support the statement that support strengthens the participants' capability.	Once again, we appreciate reviewer 2 pointing out that our main points do not come across as intended. We recognise that some statements of participants' strengthened capability can be hidden (for instance in the sentence "The participants felt an enhanced power by gaining access to self-help devices (e.g., orthoses).") However, in first submitted manuscript version we do also mention the words "capable", "capability" and "capabilities" 12 times in the findings so we do believe that it is present in the findings. We hope that by providing more quotes and rephrasing parts of the findings, we have clarified this statement.		Changes already made due to other comments.	
Line 12 This was not established in the findings which stated that pets are indifferent	As stated in our response to one of reviewer 2's comments on "being a valuable player and not just the injured one", this is due to a language barrier, and we are very thankful for having this pointed out to us as the error resulted in stating the opposite from what we wanted.		Changes already made due to other comment.	
The paragraph about complementary	We have previously discussed this paragraph within the research group and	The definition of complementary therapies is not	Text removed.	

therapies is difficult to understand. What therapies were discussed by the participants, and did they feel supported by the practitioners?	have now examined it again. Our thought was to highlight that complementary therapies (CAM) can have a role related to the meaning of support, but then we also had to address the definition of CAM. The paragraph and its focus tended to grow beyond our first intentions. With this comment from reviewer 2 in mind, we have now decided that the paragraph redirects too much emphasis on one aspect of social networks instead of keeping track of the main findings. Hence, we have removed this section.	universally agreed upon (31). In this study therapies not provided by the healthcare sector and complement regular care are considered complementary. Except for mindfulness, all therapies mentioned by the participants are listed by Wieland et al. (31) as complementary. Our findings suggest that complementary therapies can provide hope when regular care fails. This result concurs with Hsu et al. (32), who found that people with pain who use complementary therapies often have low expectations but hope they will provide pain relief and improved function, fitness, and well-being.		
You have not demonstrated that people living with chronic pain strengthen their capabilities when they accept support. More information about the capabilities approach may help to	We believe our answer to reviewer 2's first comment under "discussion" also covers this comment's first point. We used the referenced article to shed light on the findings' possible connection to person-centred care and that it can empower patients	A systematic review investigating empirical evidence underpinning the conceptualizations of person-centred care for serious illness found that person-centred	(32). A systematic review investigating empirical evidence underpinning the conceptualisations of person-centred care for serious illness found that person-centred	26/3-8

strengthen this argument. The systematic review mentioned is in the context of serious illness. Please use a reference for patient-centered care and chronic pain.	and their families. We do not know of any systematic review that states the same thing about persons living with chronic pain, but if there is such an article, we would like to know about it and consider replacing the article. We have added a sentence to clarify the reference's role in this manuscript.	care empowers patients and their families by providing information and including them in all decision-making actions on their daily lives and care (35).	care empowers patients and their families by providing information and including them in decision-making actions on their daily lives and care (33). Considering these results, person-centred care may be valuable in enhancing capability. It would be worthwhile to explore whether this is the case among persons with chronic pain and their close others.	
Methodological considerations				
You have used a reference about Naturalistic Inquiry, not phenomenology. The methodology, theoretical framework and methods used to address the research question must be coherent. Research rigour is achieved when transparency, reflexivity and positioning are made explicit. This has	We very much appreciate this comment, as we understand that adding the reference to naturalistic inquiry only created confusion instead of clarification. To address this issue we have revised the text. The revised text is largely based on a new article (from 2022) by Lindseth and Norberg. This article has replaced both the reference to Naturalistic Inquiry and in some parts the older article of Lindseth and Norberg (from 2004), which we had used previously. We believe their new article better clarifies assumptions related to trustworthiness and this may also show in the revised manuscript text. However, transparency, reflexivity and positioning are not terms	This study has several limitations and strengths. One way to determine trustworthiness in qualitative research is through credibility, dependability, transferability, and conformability (35). Lindseth and Norberg state that phenomenologica I hermeneutics seeks not to enclose the whole truth but to present meanings of a lived	Please see the section “methodological considerations.”	26/10-23 And 27/1-23

not been achieved in this manuscript.	used in phenomenological hermeneutics as described by Lindseth and Norberg. They describe other concepts, such as validity, reliability and universality.	phenomenon vis-à-vis interpreting the narrative text (23). Discussions between the authors ensured the interpretations were plausible while not being the only possibility and that dependability was strengthened. Using the hermeneutic spiral, pending between understanding and explanation by looking at the parts and the whole, helps ensure the credibility of the findings. The study's confirmability was considered because the pre-understanding was noted and used in the comprehensive analysis. [...] Most participants in the present study were well-educated, female, and born in Sweden, which could affect the transferability of the findings. Because all		
--	--	--	--	--

		participants had lived with pain for many years, they had broad experiences of support and provided rich data, enhancing the study's transferability. However, Bruce et al. described the chronic "pain journey to acceptance" and that different coping mechanisms are useful depending on where the person is in the journey (40), which might also translate to support. Because all participants had lived with pain for many years, they might have gotten further in their journey and the transferability to persons just starting their journey is unclear.		
--	--	--	--	--